# High and Sustained Ex Vivo Frequency but Altered Phenotype of SARS-CoV-2-Specific CD4^+^ T-Cells in an Anti-CD20-Treated Patient with Prolonged COVID-19

**DOI:** 10.3390/v14061265

**Published:** 2022-06-10

**Authors:** Leon Cords, Maximilian Knapp, Robin Woost, Sophia Schulte, Silke Kummer, Christin Ackermann, Claudia Beisel, Sven Peine, Alexandra Märta Johansson, William Wai-Hung Kwok, Thomas Günther, Nicole Fischer, Melanie Wittner, Marylyn Martina Addo, Samuel Huber, Julian Schulze zur Wiesch

**Affiliations:** 1Infectious Diseases Unit I, Department of Medicine, University Medical Center Hamburg-Eppendorf, 20246 Hamburg, Germany; l.cords@uke.de (L.C.); knapp.maximilian@gmail.com (M.K.); r.woost@uke.de (R.W.); sophia-schulte@gmx.de (S.S.); s.kummer@uke.de (S.K.); c.ackermann@uke.de (C.A.); c.beisel@uke.de (C.B.); m.wittner@uke.de (M.W.); m.addo@uke.de (M.M.A.); s.huber@uke.de (S.H.); 2German Center for Infection Research (DZIF), Partner Site Hamburg-Lübeck-Borstel-Riems, 20246 Hamburg, Germany; nfischer@uke.de; 3Institute of Transfusion Medicine, University Medical Center Hamburg-Eppendorf, 20246 Hamburg, Germany; s.peine@uke.de; 4Benaroya Research Institute at Virginia Mason, Seattle, WA 98101, USA; ajohansson@benaroyaresearch.org (A.M.J.); bkwok@benaroyaresearch.org (W.W.-H.K.); 5Leibniz Institute for Experimental Virology (HPI), 20251 Hamburg, Germany; thomas.guenther@leibniz-hpi.de; 6Institute of Medical Microbiology, Virology and Hygiene, University Medical Center Hamburg-Eppendorf, 20246 Hamburg, Germany

**Keywords:** SARS-CoV-2, COVID-19, CD4^+^ T-cells, T-cell memory, MHC class II Tetramer, PD-1, TIGIT, anti-CD20 therapy, CD39, CD73

## Abstract

Here, we longitudinally assessed the ex vivo frequency and phenotype of SARS-CoV-2 membrane protein (aa145–164) epitope-specific CD4^+^ T-cells of an anti-CD20-treated patient with prolonged viral positivity in direct comparison to an immunocompetent patient through an MHC class II DRB1*11:01 Tetramer analysis. We detected a high and stable SARS-CoV-2 membrane-specific CD4^+^ T-cell response in both patients, with higher frequencies of virus-specific CD4^+^ T-cells in the B-cell-depleted patient. However, we found an altered virus-specific CD4^+^ T-cell memory phenotype in the B-cell-depleted patient that was skewed towards late differentiated memory T-cells, as well as reduced frequencies of SARS-CoV-2-specific CD4^+^ T-cells with CD45RA^−^ CXCR5^+^ PD-1^+^ circulating T follicular helper cell (cT_FH_) phenotype. Furthermore, we observed a delayed contraction of CD127^−^ virus-specific effector cells. The expression of the co-inhibitory receptors TIGIT and LAG-3 fluctuated on the virus-specific CD4^+^ T-cells of the patient, but were associated with the inflammation markers IL-6 and CRP. Our findings indicate that, despite B-cell depletion and a lack of B-cell—T-cell interaction, a robust virus-specific CD4^+^ T-cell response can be primed that helps to control the viral replication, but which is not sufficient to fully abrogate the infection.

## 1. Introduction

Coronavirus disease 2019 (COVID-19) is a disease triggered by the Severe Acute Respiratory Syndrome Coronavirus type 2 (SARS-CoV-2). Dependent on the host’s age and presence of comorbidities, as well as the vaccination status and circulating variant, the disease severity shows a broad clinical spectrum, from little or no symptoms at all up to severe pneumonia and respiratory failure. In particular, elderly and comorbid patients were considered to be more vulnerable to severe courses in the early phases of the pandemic [1]. On the contrary, most immunosuppressants are not believed to be major risk factors for severe COVID-19 [2]. However, it is known that immunosuppression is associated with viral persistence and escape mutations [3,4,5,6,7].

Generally, protective immunity against viruses such as SARS-CoV-2 is believed to develop from the interplay of specific T- and B-cells that have distinct, but complementary functions [8,9]. The properties of T-cell immunity in COVID-19 most likely modulate the course of the disease [10,11,12]. While direct antiviral activity is typically exerted by cytotoxic CD8^+^ T-cells, CD4^+^ helper T-cells concert other immune cell types [13,14]. The CD4^+^ T-cell—B-cell interaction is necessary to induce solid plasma B-cell generation, as well as build a strong antibody response [14,15,16]. On the contrary, the interactions of T- and B-cells is thought to be important for a functioning CD4^+^ T-cell memory, which allows for a more rapid immune response against a re-encountered pathogen [17,18,19,20].

A particular CD4^+^ T-cell subset that is specialized in B-cell help is the T follicular helper cell (T_FH_) subset located in the germinal centers of secondary lymphoid organs [14]. In the peripheral blood, a subset of CD4^+^ T-cells with a similar phenotype and function has been described, termed circulating T_FH_ cells (cT_FH_) [21]. In COVID-19, cT_FH_ have been proposed to play a crucial role in antiviral immunity [22]. SARS-CoV-2-specific cT_FH_ are expanded in convalescent individuals, and are associated with antibody titers [23,24,25].

While SARS-CoV-2-specific CD8^+^ T-cells have already been phenotypically studied in this context [26,27,28,29], SARS-CoV-2-specific CD4^+^ T-cells have rarely been studied through major histocompatibility complex (MHC) multimer staining [30,31,32]. In particular, we lack data on the longitudinal ex vivo frequency and phenotype over the course of the disease.

Currently, several B-cell-depleting anti-CD20 immunotherapies are approved for certain autoimmunity-related conditions and hematologic malignancies. In these cases, the depletion of B-cells often results in reliable disease outcomes [33,34,35,36,37]. However, those same therapies do not only affect the anti-tumor response, but also the immune response to pathogens and vaccines [38,39,40,41].

Here, we present a longitudinal analysis of the SARS-CoV-2-specific CD4^+^ T-cell response in two COVID-19 patients. We investigated the frequency and phenotype through MHC class II Tetramer staining in a B-cell-depleted patient with a prolonged course of COVID-19 and in a fully immunocompetent patient over the course of 391 and 159 days, respectively.

## 2. Materials and Methods

### 2.1. Patients and Sample Processing

Peripheral blood mononuclear cells (PBMC) from both patients and healthy controls were collected at the University Medical Center Hamburg-Eppendorf, isolated by density gradient centrifugation, and stored at −196 °C in nitrogen. SARS-CoV-2 infection was confirmed as previously described by RT–PCR from nasopharyngeal swabs [42]. Viral isolation and next-generation sequencing (NGS) from a sample of the index patient were performed as previously described [43]. Relevant laboratory results were extracted from the clinical database. HLA typing was performed at the Institute of Transfusion Medicine at the University Medical Center Hamburg-Eppendorf using a PCR-sequence-specific oligonucleotide (PCR–SSO) with the commercial kit SSO LabType (One Lambda, Canoga Park, CA, USA), as described previously [44]. The patients gave written informed consent, and the study was approved by the Institutional Review Board of the Ärztekammer Hamburg (PV7298, PV4780).

### 2.2. MHC Class II Tetramer Staining

Cryopreserved PBMC were thawed in RPMI1640 medium (Gibco, Thermo Fisher Scientific, Waltham, MA, USA) and washed. A PE-labelled DRB1*11:01 MHC class II Tetramer loaded with a 20-mer sequence of the SARS-CoV-2 membrane protein (aa145–164; LRGHLRIAGHHLGRCDIKDL) was added at a final concentration of 2.5 µg/mL to the cell suspension of 1 × 10^7^ cells/mL for 30 min at room temperature [20]. Subsequently, the cells were washed and stained with a 1:150 dilution Zombie NIR fixable viability dye (BioLegend, San Diego, CA, USA) and monoclonal antibodies, as indicated in Appendix A, for 30 min at room temperature, and fixed with 4% paraformaldehyde for 45 min at 4 °C.

### 2.3. Intracellular Cytokine Staining (ICS)

Cryopreserved PMBCs were thawed in RPMI1640 medium (Gibco) and washed. 1 × 10^6^ cells were cultured in R10 medium (RPMI1640 supplemented with 10% heat-inactivated fetal calf serum, 1% HEPES (Gibco) and 1% penicillin/streptomycin). The cells were cultured for 18 h with 20 µg/mL of PE-labelled anti-CD107a (clone H4A3, BioLegend) and a peptide pool of 10 non-spike 15-mer peptides (Appendix A) [30] at 10 µg/mL at 37 °C and 5% CO_2_. After 1h, Brefeldin A (Sigma-Aldrich, St. Louis, MO, USA) was added in a final concentration of 5 µg/mL. Under the same conditions, SEB (1 µg/mL) served as a positive control, and solely R10 medium as a negative control. Subsequently, the cells were washed and stained with a 1:150 dilution Zombie NIR fixable viability dye (BioLegend) and monoclonal antibodies targeting surface antigens (Appendix A) for 30 min at room temperature. The cells were fixed and permeabilized with the Foxp3 transcription factor staining buffer kit (eBioscience, Thermo Fisher Scientific, Waltham, MA, USA) for 45 min at 4 °C, and stained for intracellular cytokines (Appendix A) with monoclonal antibodies.

### 2.4. Ex Vivo IFN-γ ELISpot Assay

Freshly isolated PBMC were plated on pre-coated IFN-γ ELISpot plates (IFN-γ ELISpot PLUS kit; Mabtech AB, Nacka Strand, Sweden) at 250,000 cells/well. The cells were incubated with a pool of 10 non-spike 15-mer peptides (Appendix A) [30] at 10 µg/mL for 24 h at 37 °C and 5% CO_2_. Under the same conditions, an anti-CD3 stimulation, as provided by the manufacturer (mAb CD3-2; 1:1000 dilution), served as a positive control, and solely R10 medium served as a negative control.

### 2.5. Data Analysis

For flow cytometric analysis, cells were acquired on the same day on a BD–LSR Fortessa flow cytometer (BD Biosciences, Heidelberg, Germany) run by BD FACSDiva software v8.0 (BD Biosciences, San Jose, CA, USA). Data analysis was performed in FlowJo v10.7 (FlowJo LLC, Ashland, OR, USA) for Windows. Graphs and plots were created using GraphPad Prism v7.05 (GraphPad Software Inc., San Diego, CA, USA) for Windows and NIH SPICE v6.1 (NIH NIAID, Bethesda, MD, USA) for MacOS [45]. Data are depicted as mean with standard deviation. The following analyses were performed: for statistical testing, Mann–Whitney U test; for correlation analysis, Pearson correlation. Results were considered statistically significant if *p* < 0.05. Levels of significance are translated to asterisks as follows: ns *p* ≥ 0.05; * *p* < 0.05; ** *p* < 0.01.

## 3. Results

### 3.1. Clinical Course

The clinical course and the therapeutic approach of this specific B-cell-depleted patient have been previously described by Malsy et al. [46]. The patient is a 53-year-old female, who has been treated for follicular lymphoma with a CHOP chemotherapy regimen (cyclophosphamide, hydroxydaunorubicin, oncovin, prednisone) and an anti-CD20 monoclonal antibody (Obinutuzumab) for maintenance therapy in 8-week intervals. She received anti-CD20 therapy in January 2020 for the last time. Early in March 2020, the patient became infected with SARS-CoV-2, most likely on a holiday trip to Austria. In mid-March, she began to develop fever, myalgias, asthenia, a dry cough, and mild dyspnea [46]. She tested positive for SARS-CoV-2 via PCR from a nasopharyngeal swab three days afterward. Her relevant laboratory results are depicted in Figure 1A. The B-cells were not detectable during the whole period studied, and the T-cell counts were strongly reduced, except for a short period around day 120. Due to recurrent dyspnea with peripheral oxygen saturation < 90%, her course of COVID-19 was classified as severe, according to the STAKOB [47]. Concomitant with the recurring disease, we could observe at least two peaks of the inflammatory markers C-reactive protein (CRP) and interleukin-6 (IL-6) (Figure 1A). Viral clearance from nasopharyngeal material (defined as two consecutive negative test results) occurred spontaneously 23 days after the onset of symptoms. However, due to recurrent symptoms and persistent positive results from her sputum samples, she was treated with remdesivir and convalescent plasma [46].

We present a 50-year-old woman as a sex- and age-matched, fully immunocompetent reference patient with a mild course of COVID-19 who was also positive for the DRB1*11 allele. After having contact with an infected colleague at her workplace in late October 2020 [48,49], she went into quarantine, and began to develop symptoms (headache, fever, myalgia, rhinitis) four days later. She tested positive for SARS-CoV-2 via PCR from a nasopharyngeal swab two days after the onset of symptoms. Her mild course of disease made an admission to the hospital unnecessary. Thus, no laboratory results are available. Spontaneous viral clearance was achieved 21 days after the onset of symptoms.

The patient characteristics and HLA-type are summarized in Table 1.

### 3.2. Delayed Contraction Phase of SARS-CoV-2-Specific Effector CD4^+^ T-Cells

The frequency and phenotype of SARS-CoV-2-specific CD4^+^ T-cells targeting an HLA-DRB1*11 restricted epitope (LRGHLRIAGHHLGRCDIKDL) of the SARS-CoV-2 membrane protein were assessed via ex vivo MHC class II Tetramer staining of PBMC [30]. In Figure 1B, an exemplary flow cytometry plot is shown, and the full gating strategy is reproduced in Appendix A. During the entire period investigated, the reference patient showed higher frequencies of SARS-CoV-2-specific CD4^+^ T-cells. With a mean of 0.163% (range 0.047–0.453%) of all CD4^+^ T-cells in the reference patient, we could detect a roughly 7.5-fold higher frequency (*p* = 0.0043) of Tetramer^+^ CD4^+^ T-cells, as compared to the reference patient (mean 0.022%, range 0.006–0.04%; Figure 1C).

Next, we analyzed the composition of memory subsets within the epitope-specific CD4^+^ T-cells. Generally, within the virus-specific CD4^+^ T-cells, the index patient showed a higher proportion of late-differentiated CCR7^−^ CD45RA^−^ effector memory (T_em_; *p* = 0.017) and CCR7^−^ CD45RA^+^ late effector memory (T_emRA_; *p* > 0.05) cells, whereas the reference patient rather showed an early-differentiated CCR7^+^ CD45RA^−^ central memory (T_cm_; *p* > 0.05) subset (Figure 1D,E and Appendix A). One exemplary blood sample of each patient obtained at a similar time after the onset of symptoms (day 32 for the index patient, day 34 for the reference patient; Appendix A) confirms this observation: while in the index patient with 65.1% of all epitope-specific CD4^+^ T-cells a T_em_-dominated immune response could be seen, the T_em_ subset only represented 21.2% in the reference patient. The reference patient instead presented with a 55.9% prominent T_cm_-subset, as compared to 19.5% in the index patient.

At the same time, terminally differentiated T_emRA_ cells could be detected in the index patient (7.86%), but not in the reference patient. The longitudinal assessment of the SARS-CoV-2-specific CD4^+^ T-cell memory phenotype is depicted in Appendix A. Despite this skewed memory phenotype, the virus-specific cytokine responses of the CD4^+^ T-cells of the index patient after in vitro stimulation with a non-spike peptide pool (SARS-CoV-2 M, N, and E-proteins) had a higher magnitude and greater polyfunctionality, as compared to the reference patient (Appendix A).

To better understand the kinetics and quality of the SARS-CoV-2-specific CD4^+^ T- cell response, we also investigated the expression of CD127 (IL-7Rα) and compared it to the bulk CD4^+^ T-cells. Initially, the frequency of CD127^+^ virus-specific CD4^+^ T-cells was reduced, compared with the bulk CD4^+^ T-cells of the two patients (Appendix A) and healthy controls (median 89.7%, IQR 84.6–91.55%). Only after spontaneous (reference patient HH-20-225) and treatment-induced (index patient HH-20-044) resolution and control of the infection did the proportion of CD127^+^ virus-specific CD4^+^ T-cells increase to the levels of the bulk CD4^+^ T-cells. The contraction of CD127^−^ virus-specific effector cells appeared to occur later in the index patient (Figure 1F). Even after the contraction phase, we could still detect a relevant IFN-γ response to a pool of non-spike peptides [30] in an ex vivo ELISpot assay (Appendix A) 8 months and 17 months after the resolution of the virus in the reference and the index patient, respectively.

We also investigated the frequency of T-cells with a CD45RA^−^ CXCR5^+^ PD-1^+^ cT_FH_ phenotype. With a mean of 0.83% of the CD4^+^ T-cells, the bulk cT_FH_ cells in the index patient were only slightly reduced (*p* > 0.05), as compared to the reference patient (mean 1.3%) and healthy controls (median 2.35%, IQR 1.17–3.06%) (Appendix A). However, within the SARS-CoV-2-specific CD4^+^ T-cell compartment, the reference patient showed a markedly reduced frequency of cells with a cT_FH_ phenotype (Figure 1G and Appendix A). While in the reference patient, a mean of 18.56% of the Tetramer^+^ CD4^+^ T-cells were CD45RA^−^ CXCR5^+^ PD-1^+^, in the index patient, these cells only made up 3.79% (*p* = 0.0043).

In summary, we observed a slightly higher but stable frequency of SARS-CoV-2-specific CD4^+^ T-cells in the B-cell-depleted patient, as compared to an immunocompetent individual. The memory phenotype of these virus-specific CD4^+^ T-cells was skewed towards late-differentiated subsets, and the contraction of effector cells appeared to be delayed. In the index patient, only a very low percentage of the virus-specific T-cells showed a cT_FH_ phenotype.

### 3.3. Co-Inhibitory Receptor Expression of SARS-CoV-2-Specific CD4^+^ T-Cells Correlates with Inflammation Levels

We analyzed the expression patterns of the co-inhibitory receptors programmed cell death protein 1 (PD-1), T-cell immunoreceptor with Ig and ITIM domains (TIGIT), and lymphocyte-activation gene 3 (LAG-3) on bulk and virus-specific CD4^+^ T-cells, and assessed associations between different markers and phenotypes on the virus-specific CD4^+^ T-cells (Appendix A). For all three co-inhibitory receptors and in both patients, we could consistently observe higher frequencies in the specific CD4^+^ T-cell compartment, as opposed to the bulk CD4^+^ T-cells (Appendix A). The expression of each co-inhibitory receptor developed distinctly throughout the infection.

The frequency of PD-1^+^ Tetramer^+^ T-cells was notably higher in the index patient than in the reference patient (Figure 2A). Between the measured points on day 32 and day 112 after the onset of symptoms, the PD-1^+^ frequencies were around 2-fold higher (maximum 92.3% of specific CD4^+^ T-cells on day 112) than at the other time points in the index patient. The reference patient did not show a similar episode of such high frequencies (maximum of 72.4% of specific CD4^+^ T-cells on day 25). Of note, the frequencies of PD-1^+^ Tetramer^+^ T-cells correlated with CD127^−^ Tetramer^+^ T-cells for both patients (r = −0.774; *p* = 0.005).

As described previously [50], an increase in LAG-3^+^ bulk CD4^+^ T-cells to a maximum of 2.77% (index patient, day 32) and 2.64% (reference patient, day 4), compared to healthy individuals (median 1.07%, IQR 0.86–1.62%), could be observed during the acute infection (Appendix A). LAG-3^+^ frequencies decreased again afterward. With 28.9% and 27.8%, respectively, a more marked increase was visible in the specific CD4^+^ T-cells of both the index and the reference patient (Figure 2B). For the index patient, the frequency of virus-specific LAG-3^+^ CD4^+^ T-cells showed a non-significant association (r = 0.945; *p* = 0.055) with the available CRP levels.

The specific CD4^+^ T-cells of both patients showed a slightly higher TIGIT^+^ frequency, as opposed to the bulk CD4^+^ T-cells (Appendix A). The frequencies of TIGIT^+^ Tetramer^+^ CD4^+^ T-cells in the index patient were lower, as compared to the reference patient (Figure 2C). The longitudinal sequence did not seem to follow any specific order at a first glance, but the TIGIT^+^ frequencies on both virus-specific (r = 0.997; *p* = 0.003) and bulk (r = 0.966; *p* = 0.034) CD4^+^ T-cells of the index patient could be significantly correlated with the available plasma IL-6 levels.

When assessing the co-expression of the three co-inhibitory receptors on the specific CD4^+^ T-cells, we noticed steady proportions in the reference patient (Appendix A). The SPICE analysis on day 34 revealed a dominant co-expression of PD-1 and TIGIT (Appendix A). In contrast, the amount of co-expressed receptors varied more in the index patient. Notably, between days 32 and 112 after the onset of symptoms, the proportion of cells expressing no inhibitory receptor decreased drastically. On day 32, PD-1 and LAG-3 were preferentially expressed on the virus-specific CD4^+^ T-cells of the index patient.

Taken together, we observed increased frequencies of co-inhibitor receptors on virus-specific CD4^+^ T-cells of both patients. For the B-cell-depleted patient, the expression of LAG-3 and TIGIT on virus-specific CD4^+^ T-cells was associated with the degree of inflammation.

### 3.4. CD39 Expression Marks SARS-CoV-2-Specific CD4^+^ T-Cells in the Acute Infection

Increasing evidence has been generated for the immunomodulatory functions of purinergic signaling in viral infections such as HIV and other infectious diseases [51,52,53,54,55,56]. During inflammation and tissue damage, adenosine triphosphate (ATP) is released to extracellular compartments, and acts as a pro-inflammatory mediator [57,58,59]. Two important actors of ATP metabolism are the ectoenzymes ectonucleotide triphosphate diphosphohydrolase 1 (ENTPD1; CD39) and ecto-5′-nucleotidase (E5NT; CD73). CD39 and CD73 degrade extracellular ATP, and convert it to anti-inflammatory adenosine [60,61]. Furthermore, CD39 is highly expressed on CD4^+^ regulatory T-cells [62,63]. The CD39/CD73 axis has previously been described in SARS-CoV-2 infection [56]. We were interested in the expression pattern of CD39 and CD73 on antigen-specific CD4^+^ T-cells in SARS-CoV-2 infection. Representative flow cytometry plots are depicted in Figure 3A,B.

With a mean of 12.31% (range 9.01–15.7%) of bulk CD4^+^ T-cells in the index patient expressing CD39, the proportion of CD39^+^ CD4^+^ T-cells was increased, as compared to the reference patient (mean 7.69%, range 7.02–8.28%) and healthy individuals (median 5.38%, IQR 5.21–7.67%). However, the increased expression was even higher in the virus-specific CD4^+^ T-cells (Figure 3C). CD39^+^ virus-specific CD4^+^ T-cells increased during infection, peaked at day 88 after the onset of symptoms (63.2%) for the index patient and day 25 (41.38%) for the reference patient, and continuously decreased afterward.

The expression of CD73 also differed markedly on bulk CD4^+^ T-cells. The index patient showed around a 2-fold lower frequency of CD73^+^ CD4^+^ T-cells (mean 16.93%, range 10.1–21.8%), as compared to the reference patient (mean 36.16%, range 32.6–38.8%) (Figure 3D) and healthy controls (median 32.7%, IQR 23.2–41.4%). Despite this, the general trend of the CD73^+^ subset of the SARS-CoV-2-specific CD4^+^ T-cells in the index patient closely resembled the trend in the reference patient. The frequency of CD73^+^ SARS-CoV-2-specific CD4^+^ T-cells initially decreased from levels similar to the bulk CD4^+^ T-cells to a minimum of 4.17% in the index patient and 12.42% in the reference patient. Towards the later stages of the disease, the frequency of CD73^+^ virus-specific CD4^+^ T-cells recovered to levels similar to the bulk CD4^+^ T-cell compartment. Generally, SARS-CoV-2-specific CD4^+^ T-cells expressed CD39 instead of CD73 (Figure 3E). While the frequencies of virus-specific CD73^+^ CD4^+^ T-cells did not show any association with clinical data (data not shown), the frequency of specific CD39^+^ CD4^+^ T-cells could be significantly correlated with the overall frequency of specific CD4^+^ T-cells (r = 0.695; *p* = 0.018) and the frequency of PD-1^+^ virus-specific CD4^+^ T-cells (r = 0.725; *p* = 0.012) (Figure 3F). Despite this, the co-expression of CD39 with PD-1 was highly variable (data not shown).

Overall, CD39 was more frequently expressed on virus-specific CD4^+^ T-cells, and was associated with T-cell activation and differentiation.

## 4. Discussion

In this study, we investigated the ex vivo SARS-CoV-2-specific CD4^+^ T-cell response in a B-cell-depleted patient with a prolonged clinical course and low-level viral positivity in direct comparison to a sex- and age-matched immunocompetent individual who quickly cleared infection via MHC class II Tetramer analysis.

The generation of a robust SARS-CoV-2-specific T-cell response is important for efficient viral clearance and beneficial disease outcomes [64,65]. This is not a mechanism that should be taken for granted in B-cell-depleted patients: In murine models, B-cell depletion was shown to cause loss of T-cell memory, and resulted in an inability to clear viral infection [19,20]. However, in accordance with previous reports [39,40,41,66], we could show the preservation of anti-SARS-CoV-2 T-cell immunity in this anti-CD20-treated patient via ex vivo MHC class II Tetramer analysis. The frequencies of virus-specific CD4^+^ T-cells were even 7.5-times higher, as compared to an immunocompetent control. However, while viral clearance has been described in B-cell-depleted patients [67], they are at an increased risk of prolonged viral positivity [68]. Similarly, in the index patient described in this study, viral clearance was only observed after therapeutic antiviral therapy and the transfusion of convalescent plasma [46,69]. Further immunological studies need to further dissect correlates of viral control in severely immunosuppressed patients, such as B-cell-depleted patients, stem cell transplant recipients, and solid organ transplant patients.

The role of T-cell escape in COVID-19 and prolonged courses of the disease has not yet been fully defined. Viral isolates were obtained from the sputum of the index patient, sequenced, and assessed for mutational changes in the sense of escape mutations, as previously described [43]. Importantly, while we could detect four mutational changes within the spike protein, and one each within the nucleocapsid protein and the envelope protein, no mutations were detected in the membrane protein (data not shown). In particular, the sequence of the virus in the region of the CD4^+^ T-cell epitope we assessed via Tetramer analysis was not affected. Taking the broadly directed T-cell response of the index patient into account [30], this makes viral escape an unlikely explanation for the prolonged viremia. In other viral infections, such as HCV, viral persistence is associated with an impaired and fading CD4^+^ T-cell response that can only be fully recovered through early therapy initiation [70]. It is conceivable that the virus-specific response in the B-cell-depleted patient was only sustained due to the proactive clinical management with early antiviral therapy and transfusion of convalescent plasma [46,69].

Others could show the persistence of CD127^+^ SARS-CoV-2-specific T-cells in infected patients for several weeks, suggesting the existence of a specific T-cell memory, along with the ability of homeostatic in vivo proliferation [71]. We obtained similar phenotypic results in both patients in our study. The increasing frequencies of IL-7Rα expressing virus-specific CD4^+^ T-cells to frequencies similar to bulk CD4^+^ T-cells suggest the effective generation of T-cell memory and contraction of effector T-cells [72,73,74]. While the exact time of high CD127 expression on the virus-specific CD4^+^ memory T-cells remains unknown, it appears to occur later in the index patient, as compared to the immunocompetent control. This could be indicative of delayed T-cell memory generation, which—in addition to the impaired humoral immune response—potentially contributes to delayed viral clearance in this immunosuppressed patient [11].

Interestingly, we can observe a skewing of the virus-specific CD4^+^ T-cells towards late-differentiated memory subsets in the B-cell-depleted patient, which, at least in the case of T_emRA_ cells, are thought to have a role in persistent infections, and are related to persistent antigen exposure and antigen load [75]. This was contrary to the T_cm_-dominated phenotype in the immunocompetent patient, as well as the report by Neidleman et al., which confirmed the presence of a rather prominent T_cm_ subset among SARS-CoV-2-specific T-cells [71]. Our results suggest that cT_FH_ differentiation might be somewhat impaired in the CD4^+^ T-cell compartment of the B-cell-depleted index patient since we observed markedly reduced frequencies of CD45RA^−^ CXCR5^+^ PD-1^+^ phenotypes within the de novo primed SARS-CoV-2-specific CD4^+^ T-cells, despite the existence of mostly normal bulk frequencies. This point can only be made for the peripheral blood compartment since we cannot confirm this for lymphoid and other immunologically active tissues. However, considering the need for B-cell interaction for the T_FH_ lineage commitment of CD4^+^ T-cells [14], impaired differentiation pathways are a likely explanation for our observations. Thus, despite effective priming of a cellular immune response appearing to occur in anti-CD20 treated patients [39,40,41,66], B-cell-targeted immunotherapy could interfere with T-cell homeostasis and phenotype. The T-cell differentiation, with close attention to the T_FH_ population of B-cell-depleted patients in SARS-CoV-2 and other infectious diseases, should be addressed in larger cohort studies.

We evaluated the expression of the co-inhibitory receptors PD-1, TIGIT, and LAG-3 in this setting of prolonged antigen exposure [76]. Co-inhibitory receptor expression is associated with T-cell activation, and the co-expression of multiple inhibitory receptors is a phenotypic hallmark of dysfunctional and exhausted T-cells [50,77]. Expectedly, we could observe high proportions of virus-specific PD-1^+^ CD4^+^ T-cells concomitant with a phase of high immune activation. We observed a more pronounced increase in PD-1 expression in the B-cell-depleted patient, correlating with the more severe course of disease [64]. The inverse correlation of PD-1 with CD127 expression is indicative of reduced activation after the contraction of effector T-cells and the generation of a T-cell memory, which potentially occurs after viral clearance. In line with these observations, we were recently able to show that antigen-specific CD4^+^ T-cells in acute plasmodium falciparum malaria have a low CD127 expression, and mostly express PD-1 and TIGIT [78]. Similarly, we could detect increased LAG-3 and TIGIT expression in virus-specific CD4^+^ T-cells, as compared to bulk CD4^+^ T-cells. When assessing the co-expression of the co-inhibitory receptors, reduced frequencies of SARS-CoV-2-specific CD4^+^ T-cells without co-inhibitory receptor expression could be observed in the B-cell-depleted patient, indicative of increased T-cell exhaustion. On the contrary, in a murine model of malaria, T-cells expressing co-inhibitory receptors, such as PD-1 and LAG-3, were shown to be more functional [79].Therefore, these observations should be followed up with further functional analyses [77].

Purinergic signaling is of increasing interest in many infectious diseases [51,52,53,54,55,56]. We investigated the expression of CD39 and CD73 on bulk and SARS-CoV-2-specific CD4^+^ T-cells. In accordance with previous reports [56,80], we observed mostly higher frequencies of CD73^+^ than CD39^+^ bulk T-cells. In contrast, the virus-specific CD4^+^ T-cells during acute infection showed enrichment of CD39, as compared to CD73. The expression of CD39 on virus-specific CD4^+^ T-cells significantly correlated with the overall frequency of virus-specific CD4^+^ T-cells, as well as with the PD-1^+^ Tetramer^+^ CD4^+^ T-cells. This renders CD39 a candidate marker for activated, more differentiated, and antigen-specific CD4^+^ T-cells in SARS-CoV-2 infection, similar to the already known CD39 expression by tumor-reactive T-cells [81,82]. COVID-19 patients have been shown to experience a reduced expression of CD73 on CD4^+^ and CD8^+^ T-cells, compared to healthy controls [56,83]. Interestingly, CD73 was more frequently expressed in bulk and virus-specific CD4^+^ T-cells of the reference patient, but the importance of this observation remains elusive. While CD73 expression is a hallmark of CD8^+^ T-cells with reduced functionality in COVID-19 [56], in HIV infection, CD73 expression in CD4^+^ and CD8^+^ T-cells has been shown to be associated with elite control [84,85,86]. A loss of CD73 expression might be associated with a higher differentiation status. Future studies should approach the importance and potential prognostic relevance of CD73 expression in bulk and virus-specific T-cells.

Generally, our results have to be considered rather as hypothesis-generating, since we were only able to compare two exemplary patients. Thus, effects that arise from possible confounders cannot be identified and excluded. Specifically, we cannot evaluate the association of the differential disease severity with the observed results. There is a need for larger follow-up cohort studies with additional SARS-CoV-2 CD4^+^ and CD8^+^ Tetramer specificities. In this study, we followed up on the patients over a long period, but the sampling of blood and PCR swabs were not standardized. Therefore, the investigated time points diverge and are not entirely comparable. Additionally, we have some long intervals between assessed time points for both patients, and could not provide follow-up samples from an equally long time after infection for the reference patient. However, we provide important results as starting points for further research on immune responses in B-cell-depleted patients, and the immune response against SARS-CoV-2 in particular.

Taken together, our results confirm that effective T-cell priming can occur despite anti-CD20 treatment and fully depleted B-cell compartments [39,40,41,66]. However, we saw an altered phenotype and could observe a delayed generation of T-cell memory in this specific case, with prolonged viral positivity. Therefore, we provide the first directions for future investigations of antigen-specific CD4^+^ T-cell immunity in the context of immunosuppression and, more precisely, in anti-CD20-treated patients. Since anti-CD20 immunotherapy has become more abundant in almost all medical specialties, immune responses to emerging pathogens in these patients are a highly relevant topic for clinical practice. Future studies should focus on the functionality of the T-cell response in these patients, as well as to what degree immunosuppressed patients are protected against re-encountered pathogens despite an impaired humoral immune response. Furthermore, a high-resolution phenotypical assessment of antigen-specific T-cells from vaccine recipients and COVID-19 patients with breakthrough or multiple infections is still largely missing. Potential differences in these T-cell phenotypes, which could reliably be assessed by pMHC multimer staining, could hint towards mechanisms underlying distinct disease outcomes and degrees of protection.

## Figures and Tables

**Figure 1 viruses-14-01265-f001:**
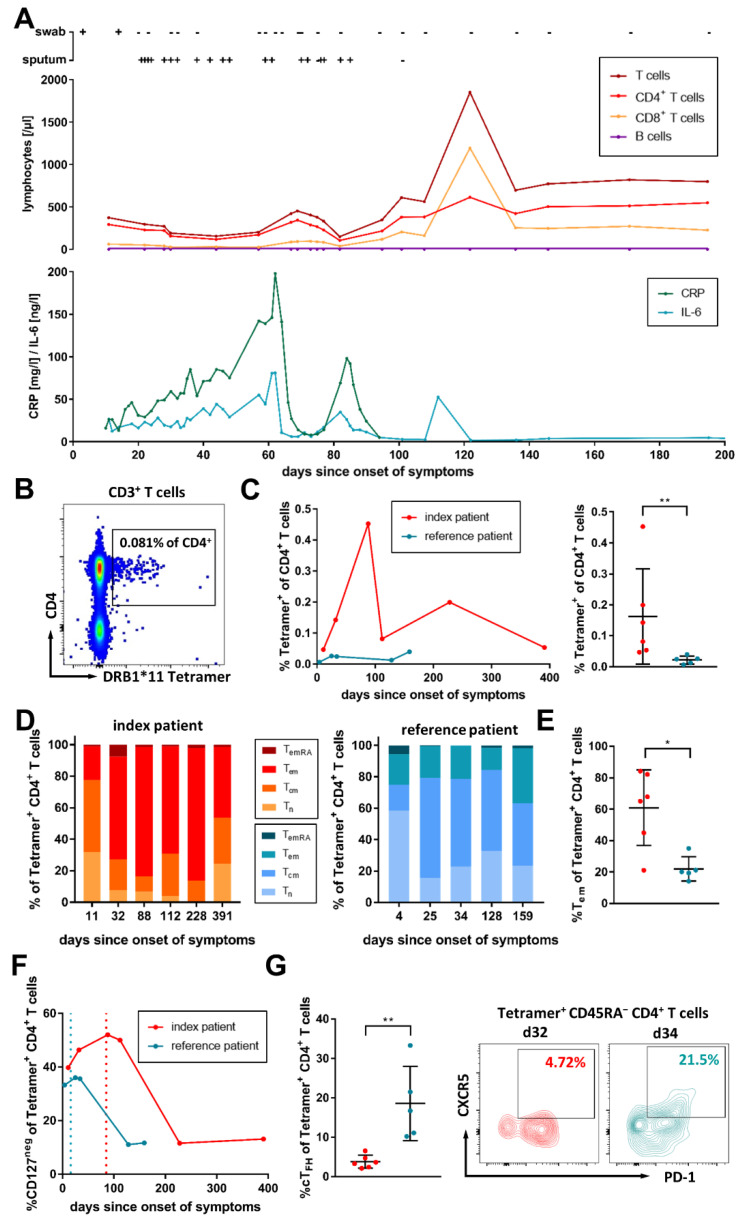
Clinical course and kinetics of Tetramer^+^ SARS-CoV-2-specific CD4^+^ T-cells. (**A**) PCR results for SARS-CoV-2 from nasopharyngeal swabs and sputum are shown as “+” for a positive and “−“ for a negative test result. Overview of the lymphocytes (B- and T-cells) and inflammatory blood markers CRP [mg/L] and IL-6 [ng/L] during the infection of the index patient are depicted. (**B**) Representative flow cytometry plot of DRB1*11 Tetramer staining of PBMC from the index patient on day 112 after the onset of symptoms. Shown are living CD3^+^ T-cells. (**C**) The frequencies of Tetramer^+^ CD4^+^ T-cells in the peripheral blood of the index patient (red) and a reference patient (blue) are depicted longitudinally (left), and pooled for each patient (right). (**D**) Comparison of the proportions of naïve (T_n_; CCR7^+^ CD45RA^+^), central memory (T_cm_; CCR7^+^ CD45RA^−^), effector memory (T_em_; CCR7^−^ CD45RA^−^), and late effector memory (T_emRA_; CCR7^−^ CD45RA^+^) T-cells between the index patient (red) and the reference patient (blue) among the SARS-CoV-2-specific T-cells. (**E**) The index patient showed cumulatively increased frequencies of SARS-CoV-2-specific CD4^+^ T-cells with a T_em_ phenotype. (**F**) Development of IL7Rα (CD127) negative SARS-CoV-2-specific effector CD4^+^ T-cells of the index (red) and the reference patient (blue). Dotted lines indicate the last positive PCR result. (**G**) The index patient showed cumulatively reduced frequencies of SARS-CoV-2-specific CD4^+^ T-cells with a circulating T follicular helper cell (cT_FH_) phenotype. Shown are representative flow cytometry plots for both patients from day 32 (index patient; red) or day 34 (reference patient; blue). In cumulative analyses, data are depicted as mean with SD, and for statistical testing, a Mann–Whitney test was performed. Results were considered statistically significant if p < 0.05. Levels of significance are translated to asterisks as follows: * *p* < 0.05; ** *p* < 0.01.

**Figure 2 viruses-14-01265-f002:**
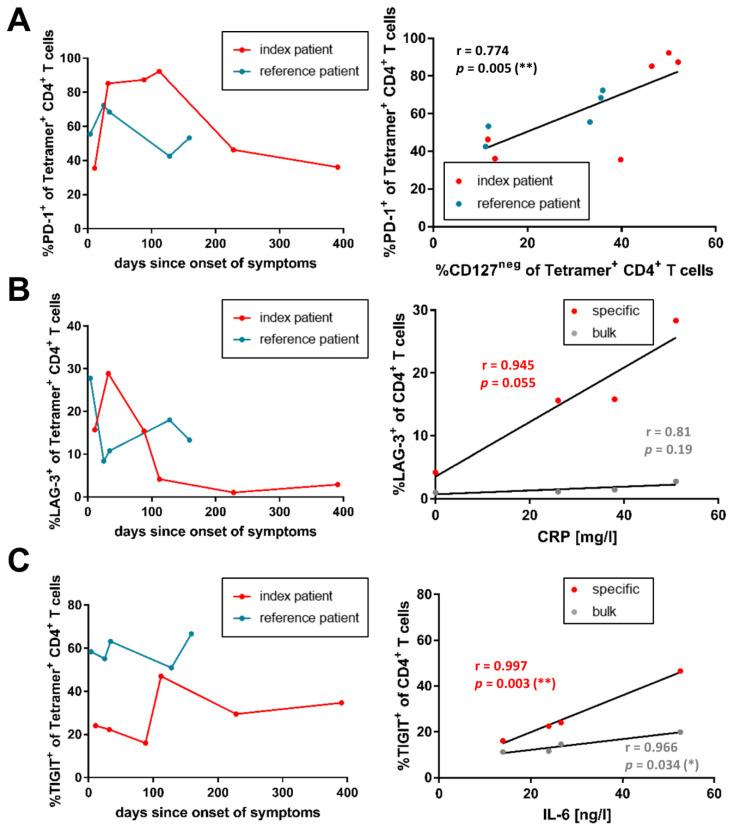
Co-inhibitory receptors on SARS-CoV-2-specific CD4^+^ T-cells. Expression frequencies of PD-1, LAG-3, and TIGIT on SARS-CoV-2-specific CD4^+^ T-cells of the index patient (red) and the reference patient (blue), as well as the respective relevant correlations, are depicted. (**A**) PD-1 expression was elevated on virus-specific CD4^+^ T-cells of the index patient, and correlated with CD127 expression. (**B**) LAG-3 expression on virus-specific CD4^+^ T-cells was elevated in early infection, and was associated with plasma CRP levels in the index patient (r = 0.945; *p* = 0.055). (**C**) TIGIT expression was elevated on virus-specific CD4^+^ T-cells of the reference patient, and significantly correlated with plasma IL-6 levels in the index patient (r = 0.997; *p* = 0.003). In correlation analyses, each dot represents an individual time point, and values are given as Pearsons’s r. Results were considered statistically significant if p < 0.05. Levels of significance are translated to asterisks as follows: * *p* < 0.05; ** *p* < 0.01.

**Figure 3 viruses-14-01265-f003:**
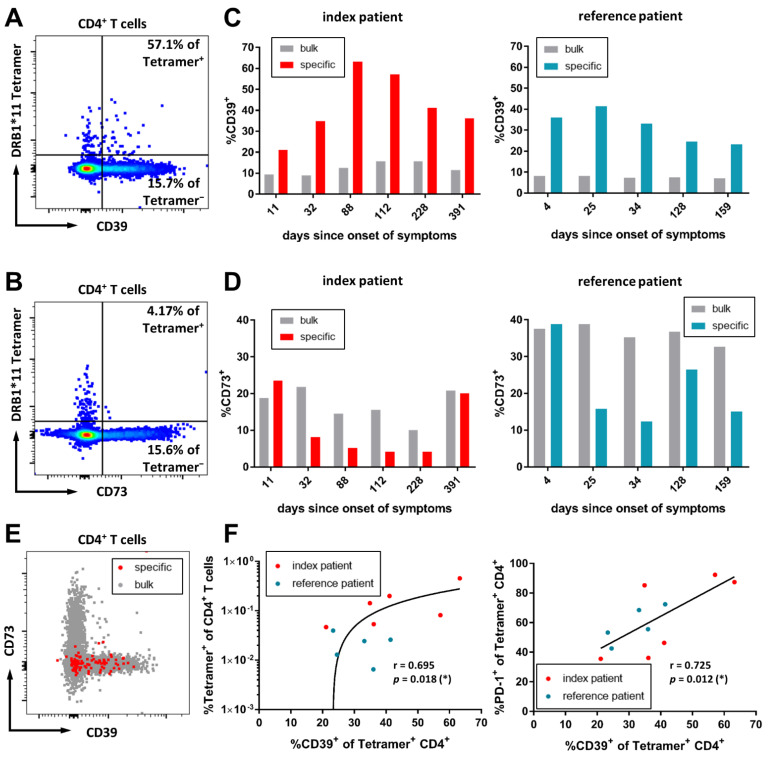
Ectonucleotidases CD39 and CD73 on SARS-CoV-2-specific CD4^+^ T-cells. Exemplary flow cytometry plots of the index patient (day 112) for expression of CD39 (**A**) and CD73 (**B**) on bulk and virus-specific CD4^+^ T-cells. Shown are living CD3^+^ CD4^+^ T-cells. (**C**) Longitudinal assessment of CD39 expression on bulk and virus-specific CD4^+^ T-cells of the index patient (left) and the reference patient (right). (**D**) Longitudinal assessment of CD73 expression on bulk and virus-specific CD4^+^ T-cells of the index patient (left) and the reference patient (right). (**E**) Representative flow cytometry plot of the reference patient, with backgating of the virus-specific CD4^+^ T-cells illustrating the CD39/CD37 expression pattern. (**F**) Correlation of CD39^+^ virus-specific T-cells with overall Tetramer^+^ CD4^+^ T-cells (r = 0.695; *p* = 0.018) and PD-1 expression frequency (r = 0.725; *p* = 0.012). In correlation analyses, each dot represents an individual time point, and values are given as Pearsons’s r. Results were considered statistically significant if *p* < 0.05. Levels of significance are translated to asterisks as follows: * *p* < 0.05; ** *p* < 0.01.

**Table 1 viruses-14-01265-t001:** Patient characteristics and HLA-type.

Patient	Sex and Age	Time of Infection	Disease Severity [47]	HLA Class II Molecules	
				*DRB1*	*DQB1*
index	female, 53	March–June 2020	severe	DRB1*07:01, ***11:01**	DQB1*02:01, *03:01
HH-20-044					
reference	female, 50	October–November 2020	mild	DRB1*03:01, ***11**	DQB1*02:01, *03:01
HH-20-225					

## Data Availability

The data presented in this study are available on request from the corresponding author. The data are not publicly available due to privacy reasons.

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
