# Peer review of "High and Sustained Ex Vivo Frequency but Altered Phenotype of SARS-CoV-2-Specific CD4+ T-Cells in an Anti-CD20-Treated Patient with Prolonged COVID-19"

_viruses, 2022, doi:10.3390/v14061265_

Round 1

Reviewer 1 Report

Reviewer #1:

Coronavirus disease 2019 (COVID-19) has emerged as a new world pandemic, infecting millions of people with a substantial mortality. There is significant interest study to analyzed the SARS-CoV-2 specific CD4+ T cells in a patient with prolonged COVID-19.

In this manuscript entitled “High and sustained ex vivo frequency but altered phenotype in 2

MHC class II tetramer analysis of SARS-CoV-2 specific CD4+ T 3 cells in an anti-CD20 treated patient with prolonged COVID-19” by Leon Cords et. Al., have conducted a highly interesting and important study to investigate the source and biological role the robust virus-specific CD4+ T cell response in an immunocompetent patient by an MHC class II DRB*11:01 tetramer analysis.

This manuscript is well written and sites key findings in the field, therefore it will be helpful for immunological investigators entering into coronavirus/COVID-19 research. The study would benefit the section on information on SARS-CoV-2 viral replication and infection.

Using this methodology, some observations are being evaluated. Therefore, comments to improve the clarity of the manuscript are provided below.

Comments for the authors' consideration:

1.

  • Line 52: There are several publications in reference to this topic. Recent MHC multimer staining are available to evaluate the specificity of CD4 and viruses. However, the authors should specify perhaps "SARS-CoV-2 specific" not "virus-specific". The following references would be good to enrich this point:

  • Gangaev, A., Ketelaars, S. L., Patiwael, S., Dopler, A., Isaeva, O. I., Hoefakker, K., ... & Kvistborg, P. (2020). Profound CD8 T cell responses towards the SARS-CoV-2 ORF1ab in COVID-19 patients. Research Square.

  • Shomuradova, A. S., Vagida, M. S., Sheetikov, S. A., Zornikova, K. V., Kiryukhin, D., Titov, A., ... & Efimov, G. A. (2020). SARS-CoV-2 epitopes are recognized by a public and diverse repertoire of human T cell receptors. Immunity, 53(6).

Line 61: The generation of memory T cells is not only dependent on B cells. I understand the context the authors: the deletion of B cells has an effect on the immune response, but it seems risky to me to mention "especially seen from the angle that B cell – T cell interactions are thought to be important..." I believe that the importance of the B-T interaction is the opposite, that is, the B cell response is the one that is dependent on T cells.  It would be good to add the references that support your perspective.

Line 69: In material and methods, to add extra information about the patient characteristics. Could be interesting some details or criteria for selecting the immunocompetent individual and not only age and gender.

Please add statistical analysis in all figures.

Reviewer 2 Report

Review, Cords et al., Viruses, 2022: High and sustained ex vivo frequency but altered phenotype in MHC class II tetramer analysis of SARS-CoV-2 specific CD4+ T cells in an anti-CD20 treated patient with prolonged COVID-19

Summary

In the study by Cords et al, the group proposed to study longitudinally the frequency and phenotype of SARS-CoV-2 epitope-specific CD4+ T cells from one anti-CD20 treated individual and from one immunocompetent donor by MHC class II tetramer analysis.

They observed higher and more stable SARS-CoV-2-specific CD4+ T cell responses in the B cell depleted donor than in the immunocompetent individual. Additionally, they found an altered virus-specific CD4+ T cell memory phenotype in the B cell depleted donor associated with inflammatory markers. In conclusion, they showed the existence of SARS-CoV-2-specific CD4 T cell responses even if B cells are depleted by treatment.

Overall, this manuscript offers interesting data and it is well written. However, some comments should to be taken into account before publication. Some important controls are missing and the conclusions are too general, based on a single donor. Authors should be clearer and more nuanced based on this unique donor.

Major comments

Introduction

  1. Authors focused on T and B cells interaction, but never introduced the T follicular helper (Tfh) population, which is of particular interest, as it provides help for B cell maturation and development of high affinity antibody (Ab) responses in the germinal center (GC) of secondary lymphoid organs. Studies have shown that a subset of CXCR5+ in blood, called circulating Tfh (cTfh) (Crotty, 2014; Morita et al., 2011) has clonal, phenotypic and functional overlap with GC Tfh and reflect at least in part responses in tissues (Heit et al., 2017; Vella et al., 2019). This cell population should be introduced here and studied throughout the manuscript. In the flow cytometry panel, authors can certainly study this population without doing new experiments because they have already included phenotypic markers for Tfh characterization (as CD45RA, CXCR5, PD1).
  2. It should be also of interest to introduce the different mechanisms by which B cell activation occurs depending on the molecular class of the antigen (T cell-independent and T-cell dependent activation of B cells)
  3. Authors should avoid redundancies in the paragraph lines 56 to 63.

Results

  1. As a first step, it should be interesting to see whether or not the anti-CD20 treated donor has a total depletion of B cell population. All your results are based on the comparison between this donor an immunocompetent donor but you have never shown if this treatment completely avoids the generation of B cells or what B cell populations are looking for. Other missing controls are an anti-CD20 uninfected and uninfected immunocompetent donor. I understand the first can be difficult to add but the second is easier to add to your analysis.
  2. Figure 1A, authors should show in parallel same information for the reference woman.
  3. Figure S1, can the authors explain why they decided to use this gating strategy? Usually, we first focus on the lymphocyte gate than single cells than DUMP-. Here, authors lose a lot of cells in their DUMP- gate. Then for CD4+ T cells characterization a double gating CD3 vs. CD4 allow the clear discrimination of this population. Moreover, on the DUMP gate it seems there is a compensation issue. Fluorochromes used have to be indicated on label’s axis for all the gating strategies shown as well as the units.
  4. Figure 1B, authors compare the frequency of SARS-CoV-2 specific T cells between the two donors but here it’s a longitudinal representation, it doesn’t give information on the comparison. Authors must adapt the representation used according to the message they want to deliver. Same comment for the figure 1D and S2A. It is very difficult to have the conclusions without reading the text. Usually, readers can be able to see the main message of each figure without report to the text. Here, it is impossible.
  5. Authors should introduce the rational of performing an ELISPOT. I don’t understand why they wanted to performed it and why the Figure S3 is in supplementary.
  6. Line 234, authors mention the CD39/CD73 axis but they didn’t develop the importance of studying it in they work. This part deserves to be further developed.
  7. Authors showed some correlations between CD39 and PD-1 but is not clear if they observed others associations. They should perform correlogram with all information they have.
  8. Figure S2B, the populations of Naïve T cells in the total population appear to be larger than others in both donors, which is not common. Have you compared with an uninfected donor? Can you show the proportion of these populations? Have you noticed a specific enrichment in this population? These results show the importance of having controls.

Discussion

  1. Authors should add a paragraph “Limitations of the study” on the fact that they study only 1 vs. 1 donor.
  2. Authors add a paragraph (line 349 to 352) on SARS-CoV-2 specific CD8 T cells, this is interesting but not well deserved, they need to improve this part to better match with the finding on CD4+ T cells done and on that particular donor.

Minor comments

  1. Line 127, authors should present markers in the same order than presented in the figure
  2. Line 129, + and – are not visible on the figure, authors should increase the size
  3. Line 133, Naïve T cells acronym is TN. Authors should mention it in the manuscript.
  4. Line 204, this are not the good values indicated on the figure
  5. Line 215, this is not the good p-value indicated on the figure
  6. Figure 3 A and B, we only see bulb PBMC not virus-specific CD4+ T cells on those figures which are in pseudocolors. Authors should represent these results as Figure S2B.
  7. Line 311, avoir repetition “to have been described”
  8. Line 339 and 375, “anti-gen” replaces by antigen
  9. Line 356, this is the inverse no, between CD73+ and CD39+
  10. Line 357, it should be better to talk about an enrichment instead of a higher expression because authors showed a frequency in a population and not the MFI. Something which could be also very interesting.
  11. Figure S2, disposition of the panels is not easy to read, start by A then next B and after C.
  12. Figure 3F, add a linear regression line instead of the curve presented.

Reviewer 3 Report

Cords L. et al studied a SARS-CoV-2 infected patient with a prolonged infection and anti-CD20 treatment, comparing with an immunocompetent SARS-CoV-2 infected patient. They provided a description of the frequency and phenotype of SARS-CoV-2-specific CD4+ T cells and the association with inflammatory markers in this patient using a MHC class II tetramer. These data contributes to a better characterization of virus-specific CD4+ T cell response in immunocompromised SARS-CoV-2 infected patients.

Major comments:

  • The clinical and T cell phenotypical characterization of the anti-CD20 patient was done in numerous time points, this could be interesting data for a better understanding of the course of the infection of these kind of patients. However, the authors should have studied sputum samples for a longer period of time, taking to account that only one negative result was obtained at day 100. In fact, there is a noteworthy peak in lymphocyte frequencies around day 120 that authors should mention and discuss, and it might be important to be sure that the patient is seronegative in this time point.
  • The authors provided a good phenotypical characterization of virus-specific CD4+ T cells, nevertheless, cytokine production and degranulation or cytotoxic capacity of these cells should be also measured and associate with observed phenotypical alterations. In fact, in addition to IFN production, other cytokines and degranulation are known to be important during SARS-CoV-2 infection.
  • An appropriate reference COVID-19 patient is a key factor for the study. The two patients are matched by sex and age. However, the anti-CD20 patient showed a severe symptomatology while the reference patient showed a mild one. This is very important since some of the differences observed in the study between the two patients might be related to a difference in disease severity. Moreover, the reference patient should have been studied for a longer period of time to compare with the index patient. The author should discuss this limitation in the discussion section.
  • The figures are not well described. All data from the figures should be mentioned in the results section, for example, the data of lymphocyte numbers and inflammatory markers in Figure 1A. The legends of the supplementary figures should be explained, the authors should write these legends as the ones from the main figures. For example, what is representing each well in Figure S3? What is Figure S3C?.
  • The authors should explain better the methods. Specify the time of incubation, temperature and concentrations. For example, for monoclonal antibody staining, fixation with PFA, anti-CD3 simulation…
  • In general, I suggest that the authors should explain better the the figures in the results section and reduce the discussion.

Minor comments:

  • The title is very long, I suggest to focus on the main message and to reduce it.
  • For the ex-vivo IFN ELISpot assay, a pool of 10 non-spike peptides were used. What protein do these peptides belong to?
  • The authors should describe how the index patient was treated with anti-CD20 (time).
  • The symptoms of the two patients should be also described. How were the patients classified as severe or mild?
  • Line 146-147. The authors should describe these results in percentages, as it is shown in the figure, to avoid confusions, and they should specify in which days higher percentages of tetramer+ CD4 T cells were found.
  • Figure 1C. Although it is described in the figure legend, a legend or a figure title within the figure might be helpful indicating which graphs represent the data from the index patient and the reference patient.
  • Line 164. Results from the day 100 shouldn’t be consider the early phase of the infection.
  • Line 167-168. Where is the figure showing that the proportion of CD127+ virus-specific cells increased to the levels of total CD4 T cells?
  • Figure 2A-C, right. Taking to account that during the study only two patients are compared, how where the correlations done? What is representing each dot? Which statistic test was used? The same for the Figure 3F.
  • The authors should include the percentages in the representative dot plot graphs.

Round 2

Reviewer 2 Report

Thank you for the outstanding responses and improvements to your manuscript! Very good work!

Reviewer 3 Report

I consider that the manuscript was considerably improved. Therefore, I suggest the manuscript to be accepted in the present form.